# Current Evidence in SARS-CoV-2 mRNA Vaccines and Post-Vaccination Adverse Reports: Knowns and Unknowns

**DOI:** 10.3390/diagnostics12071555

**Published:** 2022-06-26

**Authors:** Dimitra S. Mouliou, Efthimios Dardiotis

**Affiliations:** Department of Neurology, Laboratory of Neurogenetics, University Hospital of Larissa, University of Thessaly, 41110 Larissa, Greece; ebsdar@gmail.com

**Keywords:** COVID-19, SARS-CoV-2, mRNA, vaccination, adverse, reports

## Abstract

The novel mRNA vaccinations against COVID-19 are gaining worldwide attention for their potential efficacy, as well as for the diagnosis of some post-vaccination-reported adverse reactions. In this state-of-the-art review article, we present the current evidence regarding mainly the diagnosis of spontaneous allergic reactions, the skin occurrences, the vascular, blood, endocrine and heart events, the respiratory reports, the gastrointestinal, hepatic and kidney events, the reproductive and pregnancy issues and the muscle events, as well as the ear, eye, neurologic and psychiatric events following mRNA vaccination against COVID-19. We further present some evidence regarding the mRNA strategies, we provide important information for side effects associated with the spike protein based LNP-mRNA vaccine and its adjuvants, as well as evidence for all the possible dangerous roles of the spike protein, and we discuss our expert opinion on the knowns and the unknowns towards the topic.

## 1. Introduction

In December 2019, a novel Severe Acute Respiratory Syndrome Coronavirus 2 (SARS-CoV-2) was identified in China [1]. The World Health Organization (WHO) declared Coronavirus Disease 2019 (COVID-19) as a widespread pandemic, and, nowadays, COVID-19 seems like a possibly endemic disease.

Heretofore, scientific communities have made multifarious endeavors to monitor and defeat the pandemic [2,3,4,5,6,7,8]. Since 2020, the development of various vaccine technologies and platforms to reduce the possibility of severe COVID-19 has been expedited, as the increasing infection rates of SARS-CoV-2 worldwide have stimulated international alliances to immediately organize resources to make multiple vaccines in shortened timelines [9,10]. Up to now, some vaccines against COVID-19 have ended Phase III clinical trials with positive results [11]. A few of them have been approved or authorized, while the Centers for Disease Control and Prevention (CDC) have updated their recommendations with a preference for mRNA vaccines [12].

The major SARS-CoV-2 vaccine technologies widely performed include the weakened/inactivated adenovirus, the recombinant glycoprotein and the Lipid Nanoparticle (LNP)-encapsulated mRNA (Comirnaty by BioNTech/Pfizer, Spikevax by Moderna) [13]. Risk factors for a likely severe COVID-19 show high rates in societies; thus, exclusively people with pre-existing medical conditions and elders should get immunized against COVID-19 [14]. Until now, 64% of the world population has received at least one dose of a COVID-19 vaccine, whereas booster shoots have also taken place [15]. A recent study found that vaccination acceptance was mostly influenced by internet sources, work and social life [16].

Some vaccines have shown safety and immunogenicity; Comirnaty and Spikevax showed safe vaccine profiles and conferred favorable protection percentages, and VaxZevria had an acceptable safety profile and was found to be efficacious against symptomatic COVID-19 [17,18,19]. Nevertheless, literature and other documents reveal adverse events after vaccinations against COVID-19 [20].

The aim of this state-of-the-art review is to sum up the spontaneous adverse events following mRNA vaccinations, as well as the current evidence regarding the mRNA technology against COVID-19. Finally, some knowns and unknowns regarding mRNA vaccines against COVID-19 and an expert opinion along with some hypotheses are being discussed.

## 2. mRNA Vaccination against COVID-19 and Diagnosis of Spontaneous Post-Vaccination Adverse Events

### 2.1. Common Reactogenicities

CDC recommendations for Comirnaty for people aged > 5 years are evident, yet the Pfizer/BioNTech vaccine’s label remains for individuals aged 5–15 years since the vaccine is authorized by FDA but not yet approved for this age group [21]. Spikevax is approved by the FDA and recommended by the CDC for all people aged > 18 [22]. Local and systemic reactogenicities have been reported for both current mRNA vaccines, but they can be considered as general common events occurring after most vaccines’ administrations. For the BNT162b2 vaccine, more than half of the recipients reported pain in the injection side—mostly younger recipients presenting injection-site redness or swelling, fatigue and headache, whereas fever was reported at lower rates, and systemic events were observed within the first 1–2 days after vaccination and resolved shortly thereafter [17]. The summary of the safety profile for Spikevax reveals that pain at the injection site was the most common report, followed by fatigue, headache, myalgia and arthralgia, and, in lower than half of reports, instances of chills, nausea/vomiting, axillary swelling/tenderness, fever, injection site swelling and redness, whereas, with increasing age, there was a somewhat decreasing frequency of reactogenicity occurrences, and there was a higher rate of some adverse reactions in younger ages [23].

A study showed that slightly more than half of the responders reported at least one symptom after the BNT162b2 vaccine, and the common symptoms were the previously mentioned reactivities, as well as muscle spasm, sweating, dizziness, flushing, feelings of relief, brain fogging, anorexia, decreased sleep quality, itching, tingling, diarrhea, nasal stuffiness and palpitations, whereas approximately 4/5 of them disclosed the ability to continue daily living activities [24]. Another study for the mRNA-1273 vaccine showed that almost all responders reported at least one of the previously common symptoms, while 1/4 of them were unable to perform their daily schedule [25].

Regarding the BNT162b2 vaccine, an ongoing trial found that injection-site pain after any dose was reported in 1.5% of those aged 12-to-15-years-old and in 3.4% of 16-to-25-years-old participants; headache and fatigue were seen in more than half of the older participants and antipyretic agents were slightly more frequently administered in the younger participants, whereas, finally, serious adverse events were reported in 0.6% of those aged 12–15 and in 1.7% of those aged 16–25 years [26]. Finally, the overall safety and immune response were favorably concluded; few participants in either cohort (≤0.4% through 1 month after the second dose) experienced serious adverse events, and none were considered to be vaccine-related by the researchers [26]. As for children aged 5–11 years old and taking into account that the study will continue the follow-up for 2 years for the second dose of the BNT162b2 vaccine, more vaccine recipients (3.0%) than placebo recipients (2.1%) reported adverse events that were supposed by the researchers to be related to the vaccine/placebo, but only 0.1% reported serious adverse events [27].

### 2.2. Spontaneous Adverse Reports after SARS-CoV-2 mRNA Vaccinations

In an ongoing trial, 21% of the BNT162b2 vaccine recipients reported related adverse events, a rate that is significant more than the placebo group (5%) [17]. The authors discuss that this distribution largely reflects the inclusion of transient reactogenicity events that were reported as adverse events more commonly by the vaccine rather than placebo recipients, but few participants had serious adverse events [17]. The phase 3 clinical trial for the mRNA-1273 vaccine showed a frequency of 1.5% for grade 3 adverse events and a frequency of 9% for medically attended adverse events, which are both similar to the placebo group, whereas the serious adverse events were 0.6% in both groups [19]. Generally, adverse events that were deemed as vaccine-related were reported with a rate of 8.2% [19]. The BNT162b2 vaccine’s serious adverse events were similar in the vaccine-placebo groups (0.6–0.5%), and the BNT162b2 vaccine showed fewer post-vaccination adverse events than the mRNA-1273 vaccine; nevertheless, the latter is easier to carry and store because it is less temperature-sensitive [17,20].

The mRNA-1273 vaccine data have revealed that lymphadenopathy was very common (≥1/10), anaphylaxis and hypersensitivity events’ frequencies were unknown, acute peripheral facial paralysis and hypoesthesia and swelling face events were rare (≥1/10,000 to <1/1000) and myocarditis/pericarditis were very rare, whereas the pre-clinical data report that there was a very low toxicity/genotoxicity, that carcinogenicity studies were not performed and that, additionally, there were no vaccine-related adverse effects on female fertility, pregnancy, embryo fetal or offspring development or postnatal development [23]. Lymphadenopathy was reported in all age groups for the BNT162b2 vaccine, with the research disclosing that it was not related to the vaccination [17].

A recent document summarizes all UK spontaneous reports following BNT162b2 vaccination between 9 December 2020–6 April 2022, and counts 168,927 cases of post-vaccination spontaneous events in 485,939 vaccinations, of which, 746 were fatal, and fatalities occurred mainly because of cardiac, nervous system and respiratory disorders, and infections [28]. General and metabolic disorders, injuries, infections (including SARS-CoV-2), social circumstances, surgical/medical procedures, electric devices’ issues, congenital issues and few neoplasms (i.e., leukemia), as well as events under investigation, were also included [28]. Extensive adverse reports following current mRNA vaccinations are thoroughly presented, and some current hypotheses are described below.

#### 2.2.1. Allergic Reactions

The CDC recommends individuals with a severe allergic reaction or those that have a diagnosed allergy to any ingredient in the Comirnaty or Spikevax to not get vaccinated, whereas vaccinated people who had a severe allergic reaction due to the first vaccine dose should not continue with further doses [29]. The CDC also recommends that if someone received a COVID-19 shot and experienced a red, itchy, swelling or painful rash, the next shots should be administered at the suggested intervals; rashes can appear anywhere in a week-long window, whereas vaccine providers should be informed about a ‘COVID-arm’ and perform the next dose in the opposite arm [29]. On 14–23 December 2020, the Vaccine Adverse Event Reporting System (VAERS) detected 21 cases of anaphylaxis in 1,893,360 people who had been administered a first dose of the BNT162b2 vaccine (11.1 cases per million doses); approximately two thirds of them occurred within 15 minutes of vaccination and the most common signs were urticaria, angioedema, rash and a sense of throat closure. Most of them had a documented history of allergies or allergic reactions, such as to drugs or medical products, foods and insect stings, but a few disclosed an episode of anaphylaxis in the past due to rabies and influenza A (H1N1) vaccines [30,31]. The VAERS also identified 83 cases of non-anaphylaxis allergic reactions after BNT162b2 vaccination [30,31]. CDC reported 10 anaphylaxis cases after the first administration of 4,041,396 mRNA-1273 vaccines (2.5 cases per million doses), occurring in a 20-day window before mid-January 2021 [32]. Both the mRNA-1273 and BNT162b2 vaccinations showed a strong female predominance in anaphylactic events, but, despite the fact that the vaccines were given to a higher percentage of women than males at the time of the analyses, this does not explain the substantial female predominance among anaphylactic events [33]. A study has revealed that allergic reactions after vaccination were reported more frequently in allergic subjects and, despite the BNT162b2 vaccine being less tolerated by allergic individuals compared to non-allergic individuals, the emerging adverse conditions were mild and do not preclude a vaccination success [34]. According to CDC records, 52% of patients with anaphylactic reactions to the BNT162b2 vaccine and 80% of patients with anaphylaxis to the mRNA-1273 vaccine disclosed a previous history of allergies to treatments or vaccines [33].

A German study analyzing sudden deaths occurring in a window time of slightly more than a month supported that, in cases with a short time interval between death and vaccination, allergic shock should be considered. The authors highlight that the vaccinations should be critically discussed in multimorbid or palliative patients who are in poor medical condition [35].

#### 2.2.2. Skin Conditions

‘COVID arm’ has gained worldwide attention, and is described as some delayed large injection site reactions to mRNA vaccines [36]. This condition has been reported in some cases after mRNA-1273 vaccination, and it was described as a localized erythematous rash around the injection site, but also as a generalized erythrodermic reaction or a pruritic/non-pruritic arm rash in cases with/without medical history [37]. Two cases’ rashes resolved without intervention, and two others cleared (fully and not fully) with topical steroids and/or oral antihistamines [37]. COVID arm seems to be a delayed type of hypersensitivity reaction, and its histological findings are compatible with Erythema Annulare Centrifugum (EAC) [38]. A study regarding the BNT162b2 vaccine revealed that the immunohistochemistry for the SARS-CoV-2 spike 1A9 protein was negative [38]. In addition, other skin lesions and urticaria were reported post-vaccination; the lesions were resolved in a median of 6 days, and, finally, the authors concluded that there were no severe cutaneous reactions in the research, indicating that the BNT162b2 vaccine has a low risk of adverse skin effects [39].

A study documented 414 cases with cutaneous responses following mRNA-1273 (83%) and BNT162b2 mRNA vaccinations during 12/20–2/21 and found that local injection site responses, urticarial eruptions and morbilliform eruptions were the most common delayed big local reactions, whereas second-dose recurrence occurred in 43% of the participants [40]. Pernio/chilblains, cosmetic filler reactions, zoster, herpes simplex flares and pityriasis rosea-like reactions were additional less prevalent skin reactions, and, finally, the authors suggested an unexplained immune response that may have led to these skin lesions [40]. A response mentioned that reactions in this study were not limited to the injection site, while local skin reactions at the injection site, distant skin reactions and/or more generalized adverse reactions were all reported in these case series. It was speculated that these events are caused by the vaccines’ mRNA translation product, the spike protein [41].

Cutaneous lymphocytic vasculitis following BNT162B2 vaccination was presented in a woman without comorbidities but a previous SARS-CoV-2 infection [42]. An early Rowell’s syndrome and a Leukocytoclastic Vasculitis (LV) flare following mRNA vaccination were reported in middle-aged/old women with previous medical conditions [43,44]. A possible mechanism for LV exacerbation could be similar to that proposed for the influenza vaccine, which has been linked to vessel damage because of abnormal immunological activation with vaccine-related antigens, which promotes antibody development and immune complex deposition [44].

#### 2.2.3. Vascular and Blood Conditions

A study has shown that the BNT162b2 vaccine generates a significant rise in inflammatory markers, notably after the second dose, and a transient worsening of endothelial function at 24 h that returns to baseline at 48 h; these findings support the vaccine’s short-term cardiovascular safety [45]. Two reports of de novo vasculitis following mRNA vaccination describe an old healthy woman and a younger man with treated arterial hypertension [46]. Large Vessel Vasculitis (LVV) was reported after mRNA-1273 vaccination in an old woman that complained about cephalgia, cervicalgia, ostealgia and pain in multiple large joints and muscles that started 2–3 weeks after the first and worsened after the second dose; [18F]FDG-PET/CT demonstrated increased tracer uptakes in the large arteries of the legs [47].

There exists a report about a 60-year-old woman who died from a brain hemorrhage due to Arterial Hypertension (AH) 3 days after BNT162b2 vaccination, and another concerning a previously asymptomatic older woman who collapsed 3 days after mRNA-1273 vaccination; her diagnosis was a right hemiplegia, aphasia, agnosia and thrombocytopenia. Other abnormal blood tests were negative and a CT scan showed intracranial hemorrhage in the left basal ganglia; sadly, after 9 days, the woman passed away without any improvement [48]. A study reported nine patients with diagnosed stage 3 AH a few minutes after mRNA vaccination that lasted for a month; most patients had pre-vaccination AH history but none had a hemorrhagic event, and the authors attributed AH to LNP components [48]. Conversely, there also exists a report of a woman with a history of hypotension and anasarca episodes, where she developed hypotension, tachycardia, shock and anasarca after mRNA-1273 vaccination, but improved with supportive treatment. In addition, there is another report regarding a patient with a history of syncope and seizures that developed tachycardia, status epilepticus and a cardiac arrest with pulseless electrical activity that responded to cardiopulmonary resuscitation and epinephrine; the authors recognized that their observations do not exclude the vaccine or other reasons for such flares, and, finally, they recommended a pre-vaccination IVIG prophylaxis for diagnosed SCLC patients [49].

Deep Vein Thrombosis (DVT) was revealed in a 79-year-old man after sudden lower extremity pain and swelling a few days after the first mRNA-1273 vaccine; he proceeded to the second dose, and the investigations confirmed extensive acute DVT and a concurrent acute Pulmonary Embolism (PE). After anticoagulant therapy, he was discharged home with supplementary oxygen [50]. DVT was also reported in the upper extremity of a 27-year-old female 3 days after the second mRNA-1273 vaccination, with a negative relevant thrombophilia workup, but she improved after three months’ rivaroxaban treatment [51]. Moreover, a 66-year-old woman developed DVT after her second BNT162b2 vaccination; initially, she manifested fever with chills, fatigue, malaise and muscle pain, and, the next day, she was admitted to hospital, unable to walk. She had unremarkable, normal tests; DVT was revealed in the right peroneal vein and extending up to the popliteal vein, and, eventually, she improved quickly after apixaban [52]. The UK document reports 7522 vascular events, including 22 deaths and approximately 1500 vascular hemorrhages/hematoma disorders [28].

Thrombotic Thrombocytopenic Purpura (TTP) was discussed in a case of a 69-year-old man with various comorbidities who was exhausted and short of breath, while thrombocytopenia, anemia and hemolysis with schistocytes were all found to be considerable with the condition [53]. Thrombocytopenia reports after COVID-19 mRNA vaccines do not exceed the overall Immune Thrombocytopenia (ITP) rates, and they were estimated at approximately 0.80 per million doses [54]. A systematic literature research showed that the period between vaccination and admission was ≤3 days [55]. Increased risks of hematological and vascular events that resulted in hospitalization or death were seen for brief time intervals after the first ChAdOx1 and BNT162b2 vaccine administrations, while the risks of most of these outcomes were much higher and lasted longer after SARS-CoV-2 infection than after immunization, according to a large study [56]. Vaccination-Induced Thrombotic Thrombocytopenia (VITT) is a new type of thrombocytopenia that occurs after receiving COVID-19 vaccines, and recent research has shown a mechanism similar to heparin-induced thrombocytopenia, with antibodies against the PF4–polyanion complex as the protagonist (autoimmune or atypical heparin-induced TT) [55]. The lack of PF4 antibody testing does not rule out this instance, as thromboembolic illness has been linked to both VITT and non-VITT reasons following vaccination. Nevertheless, although anti-PF4–polyanion antibodies are common—they’re found in 25–50% of patients after cardiovascular surgery, for example—heparin-induced thrombocytopenia is not, and cerebral venous sinus thrombosis or thrombi in abdominal vessels rarely occur in patients with heparin-induced thrombocytopenia; this shows that knowledge of the pathophysiology of VITT is lacking, and that the utility of assessing pathogenic anti-PF4-related antibodies in all vaccine recipients is yet to be determined [57]. Additionally, even if an immune reaction can occur because of LNP components’ cellular damage, DNA and RNA also have polyanionic properties and could form a neoantigen when bound to PF4; also, there is no cross-reactivity between anti-PF4 antibodies and the SARS-CoV-2 spike protein [58]. Moreover, patients with previous SARS-CoV-2 infection might acquire an excess of pro-coagulant factors, which can lead to thrombosis and thrombocytopenia, but, again, the pathophysiology is unknown [53]. New experimental findings suggest that thrombosis is caused by a soluble adenoviral spike protein variant that emerges from splicing events that produce significant endothelial inflammatory events and binds to ACE2-expressing endothelial cells [57].

IgG-mediated Autoimmune Haemolytic Anemia (AHA) was reported after mRNA-1273 vaccination in a case with undetected pernicious anemia and, additionally, another woman with a history of postoperative chemotherapy for lung adenocarcinoma and a history of helicobacter pylori eradication for Idiopathic Thrombocytopenic Purpura (ITP) was reported with AHA 2 weeks after BNT162b2 vaccination [59,60]. In addition, there exists another report for severe Aplastic Anemia (AA) following BNT162b2 vaccination, but it is not clear whether antibodies derived from the SARS-CoV-2 vaccine directly contributed to AA because the anti-SARS-CoV-2 antibodies remained after the patient’s pancytopenia had been ameliorated by the allogeneic HSCT [61]. A healthy 34-year-old man had blurry vision in the inferior visual field, intermittent photopsia, several retinal hemorrhages and dilated and twisted retinal arteries. The symptoms started 2 days following the first BNT162b2 vaccine administration, and the patient was diagnosed with Branch Retinal Vein Occlusion (BRVO) [62]. To date, a couple of cases of acquired hemophilia have been reported after receiving mRNA vaccines [63].

Despite the fact that lymphadenopathy was not reported as a vaccine-induced adverse event in BNT162b2 vaccine clinical trials, there were significantly more instances in the vaccine group compared to the placebo group in all age groups [64]. Due to the fact that axillary swelling was only reported as an unintentional adverse event, the true prevalence of post-vaccination lymphadenopathy may be higher, since not all individuals have clinical symptoms, and the incidence of axillary lymphadenopathy detectable in imaging is expected to be higher [64]. A recent large-cohort study evaluating axillary lymphadenopathy with a long-term follow-up of six months following COVID-19 vaccination showed that, in reality: 44% of the participants were patients, a few cases of whom had malignant biopsies; there were no new diagnoses of malignancy during the follow-up period; and the reactive lymphadenopathy took a long time to resolve, with residual lymphadenopathy found up to 43 weeks after vaccination [65]. The authors concluded that there should be no delay in the screening mammography due to recent immunization, and that the time for reactive lymphadenopathy to resolve varied [65].

#### 2.2.4. Endocrine Conditions

The first spontaneous Subacute Thyroiditis (ST) and Grave’s disease have appeared after mRNA vaccinations in women with/without medical history; vaccines’ adjuvants combining into a potential cross-reactivity between SARS-CoV2 and thyroid antigens may explain some of the autoimmune/inflammatory reactions that occur during and after SARS-CoV2 infection and immunization, according to the authors’ discussions [66]. A group of individuals with Graves’-disease-related hyperthyroidism presented with a new onset or recurrence shortly after receiving the SARS-CoV-2 mRNA vaccination at a single tertiary facility in Singapore, and few related reports exist in the current literature [67]. Some authors suggest that, although upper respiratory viruses are the most prevalent cause of subacute thyroiditis, cases have been observed with standard inactivated viral vaccines as well as live-attenuated vaccines (i.e., influenza), and, although thyroiditis has not been documented as a typical side effect, this instance shows that there is cross recognition between the coronavirus spike protein targeted by the mRNA vaccine and healthy thyroid cell antigens [68]. Graves’ disease is known as a Th1-mediated disease, and IFN-γ is crucial in promoting thyroid antigen presentation via the HLA class II and, ultimately, the synthesis of class 1 IgG, which stimulates the TSH receptor; importantly, a study showed that the immune response to the BNT162b2 vaccine was from a Th1 type with IFN-γ production. Therefore, the vaccine’s alteration of the cytokine environment could lead to a change in T-cell repertoire in favor of a Th1 population, followed by the generation of new antibodies that stimulate the TSH receptor, possibly resulting in hyperthyroidism [69]. Thyrotoxicosis has also been discussed [70]. In a recent published retrospective population-based cohort study of Hong Kong Hospital, more than 40,000 patients treated for hypothyroidism with levothyroxine were evaluated for the risk of adverse events after COVID-19 vaccination, and the results confirmed that BNT162b2 or CoronaVac vaccination was not associated with an unstable thyroid status or an increased risk of adverse outcomes among patients treated for hypothyroidism in general [71].

#### 2.2.5. Heart Conditions

An increased risk of myocarditis after mRNA-1273 vaccination is highest in younger males, and the higher risk in younger boys following the second dose has also been assessed [23]. According to a study, there were roughly 1.316 (per 10,000) additional incidences of myocarditis among men 12–19 years old in the week following the second mRNA-1273 vaccination compared to those who were not exposed; in another study, there were 1.88 (per 10,000) additional instances of myocarditis 4 weeks after the second mRNA-1273 vaccination in boys 16–24 years old compared to the unvaccinated [23]. An analysis study based on VigiBase data for cardiovascular adverse events reported for COVID-19 vaccines found that the vaccine was associated with acute myocardial infarction, cardiac arrest and circulatory problems in those aged >75 years, and that hypertension and severe hypertension, such as hypertensive emergency, were related with vaccine administration regardless of age or gender [72]. In addition, myocardial infarction was recorded in 0.03 percent of patients in the vaccine group, according to data from Moderna’s clinical studies in the FDA Briefing Document, and, similarly, the data given in the FDA Briefing Document by the Pfizer-BioNTech COVID-19 vaccine revealed one cardiac arrest, one ventricular arrhythmia and one occurrence of acute myocardial infarction in 0.02 percent of the patients [72]. The Pfizer-BioNTech Series Drug Analysis Print for their COVID-19 vaccine, as of 28 May 2021, reported a total of 2342 cardiac adverse events, including 1098 palpitation events, 466 events of tachycardia, 108 events of atrial fibrillation, 94 events of myocardial infarction, 62 events of cardiac arrest, 63 events of cardiac flutter, 32 events of sinus tachycardia, 46 events of angina pectoris, 24 events of cardiac failure, 38 events of arrhythmia and 16 events of acute myocardial infarction [72]. The UK document presents 13,054 cardiac disorders, including >6000 palpitation events and 152 deaths [28].

On June 23, 2021, the Advisory Committee on Immunization Practices (ACIP) found that the advantages of utilizing mRNA COVID-19 vaccinations under the FDA’s EUA clearly outweigh the risks in all populations, including adolescents and young adults, after examining available evidence, including that for risks of myocarditis [73]. Myocarditis/pericarditis rates among people aged 12–39 years old have been reported as 12.6 cases per million doses for the second mRNA vaccination, by the US CDC [74]. A study analyzing data from the largest health care organization in Israel in order to evaluate the safety of the BNT162b2 mRNA vaccine revealed that vaccination was strongly associated with an increased risk of myocarditis (2.7 events per 100,000 persons), and that risk was higher after SARS-CoV-2 infection [75]. Data on patients < 21 years old showed a males’ preponderance and a median age of 15.8 years—with the condition to appear mainly after the second dose, and symptoms beginning at a median of 2 days, where 18.7% required an Intensive Care Unit (ICU) but none died [76]. Nevertheless, there exists a report for a 22-year-old male who developed chest pain 5 days after his first BNT162b2 vaccination and died 7 h later; a scattered single-cell myocytes’ necrosis was immunohistochemically revealed, which was not accompanied by inflammatory infiltrates, and no evidence of microthrombosis or infection in the heart and other organs was existent death was supposed to be causally associated with the vaccine [77]. After BNT162b2 vaccinations in Israel and following myocarditis reports, the clinical specimens’ analysis showed endointerstitial edema and neutrophils that were seen in two people’s endomyocardial biopsy samples, as well as mononuclear-cell infiltrates (monocytes, macrophages and lymphocytes), but no giant cells [78]. A recent systematic review revealed that BNT162b2 vaccine recipients had fewer systemic symptoms and less left ventricular dysfunction than mRNA-1273 recipients [79]. A strange case of acute perimyocarditis and pericardial effusion 10 days following the second mRNA-1273 vaccination was also reported, and the acute care was complicated with cardiac arrest, acute kidney injury, disseminated intravascular coagulation (DIC) and hemodynamic instability, but, eventually, the patient recovered [80]. Another study revealed an increased risk of myocarditis 1–28 days following the first BNT162b2 and first/second mRNA-1273 vaccinations, and increased risks of pericarditis and cardiac arrhythmias following a positive SARS-CoV-2 test, as well as an increased risk of arrhythmia following a second mRNA-1273 vaccination [81]. The rare myocarditis associated with mRNA vaccination has garnered scientific attention, and potential mechanisms include sex-specific differences identified in both mRNA vaccination-related myocarditis and non-COVID-19 viral myocarditis, which are due to the production of autoantibodies and hormone-related variables. The mRNA immune reactivity, antibodies to the SARS-CoV-2 spike protein cross-reacting with cardiac contractile proteins, and hormonal variations are three basic ways through which mRNA vaccines may produce hyperimmunity [82]. Undeniably, immunity, as well as genetic background, age and sex can all alter these processes. However, one could hypothesize that LPN has a direct harmful effect on cardiac cells, or that it is the result of an immunological reaction to it, or that it aggregates with the mRNA strand in vaccination preparations, whereas another could argue that there have been few reports of a similar clinical picture among recipients of other non-vaccine, LPN-containing treatments up until now [83]. It is also worth noting that the population most vulnerable to vaccine-induced myocarditis, notably the younger age groups, are frequently under-represented compared to those receiving LNP-based chemotherapeutics [83].

Takotsubo Cardiomyopathy (TCM) was also reported after mRNA vaccinations in elders with/without medical history [84,85]. Although the causality of TCM triggered by mRNA vaccination cannot be proven, it appears to be the most plausible cause, especially given that symptoms begin so quickly after the vaccination, but post-vaccination adverse reports are rare and, thus, some stories of TCM, in particular, are purely anecdotal [86]. Finally, a study showed that there was no increase in the incidence of acute myocardial infarction, stroke or Pulmonary Embolism (PE) in people aged 75 years old and older 14 days after each BNT162b2 vaccination [87].

#### 2.2.6. Respiratory Conditions

Some reports have associated BNT162b2 vaccination with acute exacerbations of interstitial pneumonia, whereas, in the UK, 108 incidences of Interstitial Lung Disease (ILD) have been linked to the vaccine [88]. ILD has rarely occurred after the administration of some drugs or vaccines, such as influenza, and, also, the first two cases of ILD appearing after COVID-19 vaccination were Asians [89].

The first report of post-vaccination asthma exacerbation described an occasional smoker with a 10-year history of asthma and seasonal allergic rhinitis [90]. In September 2021, the European Academy of Allergy and Clinical Immunology (EAACI) also stated that, following vaccine administration, all persons should be observed for at least 15 min, and it is also estimated that more than 30% of the population suffers from allergic diseases; however, statistics demonstrated no increased risk for people with allergic rhinitis or asthma [91].

We previously discussed a study revealing that there was no increase in the incidence of PE in people aged 75 years old and older 14 days after each BNT162b2 vaccination, but there exist some rare reports regarding PE [87,92]. A recent post-vaccination venous thromboembolism and PE report presents a 24-year-old boy, and the authors discuss his thrombophilia screening, which showed an FVL G169A heterozygous mutation and MTHFR A1298C homozygous mutation, trying to explain the event through the inherited thrombophilia [93]. The authors conclude that patients with a history of unprovoked DVT or PE who have not been tested for genetic risk should be offered testing before vaccination, as there is presently insufficient evidence to warrant their exemption from vaccination or the use of anticoagulant prophylaxis [93].

Post-vaccination eyelids’ purpura and chilblains, herpes simplex, herpes zoster and periorbital erythema flares in cases with pre-existing medical conditions are evident in literature, and the authors discuss that, although the immunization of patients taking biologics is required, herpes virus reactivation can be more severe if biologics have started recently, and, additionally, when starting a biotherapy and immunization at the same time within a short period of time, caution is advised [94].

An initial blurred vision symptom was firstly diagnosed as granulomatous iritis with mutton-fat keratic precipitates and small iris nodules in a 34-year-old male 5 days after BNT162b2 vaccination, but the persistent fever and malaise continuation led to a further diagnosis of mediastinal and bilateral hilar small lymphadenopathy, together with a limited granular lesion in the right lung; additionally, a parotid gland biopsy showed noncaseating granulomas and proved sarcoidosis, along with uveitis [95]. In the current literature, three cases were diagnosed with sarcoidosis-like diseases-2 cases with Lofgren syndrome (acute erythema nodosum and ankle swelling, with/without mediastinal and hilar lymphadenopathy), and one patient was diagnosed with mediastinal lymphadenopathy but no uveitis [95].

#### 2.2.7. Gastrointestinal Conditions

Generally, post-vaccination gastrointestinal occurrences were reported in 10–20% of cases in the mRNA vaccination studies, and these symptoms may be interpreted as an Inflammatory Bowel Disease (IBD) flare, resulting in patient concern and un-needed testing or therapy; however, a study showed that the mRNA vaccines do not seem to increase the risk of a disease flare in the vast majority of IBD patients [96].

Post-vaccination gastrointestinal spontaneous issues have also appeared in current literature, and there exist reports for Ulcerative Colitis (UC) flares in two middle-aged women, with an abdominopelvic CT to demonstrate contiguous colitis, where one woman was discharged after mesalamine/methylprednisolone, whereas the other one was diagnosed with diffuse colitis followed by a colonoscopy abortion due to extensive colitis; finally, complications led to a total colectomy [97]. Another case of post-vaccination-occurring UC was also reported in a 36-year-old woman with a medical history of UC and autoimmune pancreatitis, and, finally, symptoms improved after steroids [98].

Moreover, regarding the post-vaccination appendicitis occurrences, some data suggest a correlation with lymphadenopathy, but it is highlighted that the infections’ status should also be reported, since a multisystem inflammatory response can present as abdominal pain and both mimic and cause appendicitis, as in COVID-19 [99]. The immunological response could possibly play a role in the development of appendicitis. Vaccinations against COVID-19 elicit strong Th1 immune responses, and there is a hypothesized relationship between an elevated Th1 immune response and a subtype of appendicitis that is more likely to necessitate surgery [99].

A 43-year-old man was diagnosed with acute necrotizing pancreatitis 10 h after his second BNT162b2 vaccination, and the man had a background of atopy with mild seasonal rhinitis, eczema and asthma, as well as a previous pancreatitis episode 10 years ago [100]. The VigiBase also records more than 150 pancreatitis post-vaccination reports, and this event was also reported as self-resolving uncomplicated acute pancreatitis [101]. However, COVID-19 extrapulmonary manifestations include pancreatitis even as a first symptom, whereas the bi-modal pattern of symptom onset supports both the cytotoxic and immune-related pathophysiology of pancreatic damage [102].

#### 2.2.8. Hepatic Conditions

Liver injury was reported in a 46-year-old man who was administered the BNT162b2 vaccine 123 days after a liver transplantation for alcohol-associated liver disease, with the authors discussing a potential vaccine toxicity [103]. Acute Liver Injury (AKI) has also been reported in combination with immune thrombocytopenic purpura after mRNA-1273 vaccination, and this co-occurrence led the authors to think that the vaccine was the most likely inciting factor [104].

In registration trials, no hepatitis was recorded, but acute autoimmune-like hepatitis with an atypical anti-mitochondrial antibody was also evident after mRNA-1273 vaccination in a 63-year old male Caucasian patient with a history of type two diabetes and ischemic heart disease, for which, he was on therapy for over 5 years [105]. A study assessing liver injury in 16 cases, 6 of whom presented chronic liver disease, presented post-BNT162b2 vaccination occurrences of drug-induced liver injury (previous non-alcoholic fatty liver disease case), compensated cirrhosis (prev. hepatitis C and autoimmune hepatitis cases), extramedullary hematopoiesis of unknown significance on prior liver biopsy, autoimmune hepatitis and prior biliary stricture after cholecystectomy, whereas only an event of autoimmune hepatitis was reported after mRNA-1273 vaccination [106]. The authors discuss that this series of patients contained retrospective and observational data without adjudication, while they suggest that the host immune response against the vaccine’s induced spike protein causes an aberrant, autoimmune-like hepatic condition in predisposed individuals; additionally, they consider it unlikely that there is a direct hepatotoxicity, but they do not fully exclude it [106]. They finally raise questions for some patients with a risk for hepatic autoimmunity: if they should undergo pre-emptive post-vaccination laboratory monitoring, and their future in booster doses [106]. An unusual case of acute cholestatic hepatitis has also been reported in a 43-year-old woman; the case had not developed autoantibodies, but the authors discussed immune-mediated hepatitis, firstly because of the timeline from vaccine to liver alteration, which may correspond with the development of the immune response, and secondly due to the steroids’ response; they also disclosed that eosinophil infiltrate was present at histology and that this feature is more common in toxic/drug-related liver injury but also in autoimmune hepatitis [107].

#### 2.2.9. Kidney/Urinary Conditions

At least six incidences of gross hematuria have been recorded in patients with a history of biopsy-proven IgA Nephropathy (IgAN), involving both mRNA vaccines, and there was a report for a case with co-occurrences of Acute Kidney Injury (AKI) with gross hematuria and IgA nephropathy [108]. A VigiBase analysis study found that BNT162b2 and mRNA-1273 show a diverse spectrum of relationships with glomerular disorders, with the former likely to be linked to nephrotic syndrome, whereas the latter raises the likelihood of acquiring IgA nephropathy for the first time or relapsing [109]. A 76-year-old man with a history of hypertension and UV-treated cutaneous mycosis fungoid was diagnosed with membranous nephropathy (MN) after receiving the BNT162b2 vaccine, and he was treated symptomatically at first, with dietary changes and renin-angiotensin system blocking, which resulted in partial nephrotic syndrome control; after the second mRNA-1273 vaccination, his oedema worsened, leading to the first report of post-vaccination MN, with the authors concluding that mRNA vaccines may have a role in MN exacerbations [110]. Furthermore, 3 days after receiving the BNT162b2 vaccine, a 22-year-old man expressed a severe nephrotic syndrome relapse due to minimal change disease, and the authors highlighted vaccinations as a general known risk for idiopathic nephrotic syndrome (INS) relapses, concluding that individuals with INS must be closely monitored after vaccination [111].

A case report of anti-GBM disease with coexistent mesangial IgA deposits presented an older woman with no significant medical history who developed fevers, anorexia, nausea and gross hematuria 2 weeks after receiving the second mRNA-1273 vaccination, and the authors concluded that the immune response to the spike protein could be related to anti-GBM production in certain cases [112]. Another rare event after BNT162b2 vaccination was an isolated renal arteritis with infarction, occurring in a 48-year-old healthy man, and the authors deemed that the event was probably associated with the vaccine-related immune-mediated disease [113]. ANCA glomerulonephritis also occurred in a middle-aged man with a history of hypertension 2 weeks after mRNA-1273 vaccination, and the authors discussed that the increased immunological response that followed possibly set off PR3 antibodies, which can be positive due to certain drugs and hematological malignancies. Malignant tumors have also been linked to ANCA glomerulonephritis, but myeloperoxidase-O titers are generally positive in medication-induced ANCA vasculitis [114]. Lastly, a report presented a spontaneous scleroderma renal crisis following mRNA vaccination, occurring in a 34-year-old woman with a significant medical history [115].

#### 2.2.10. Reproductive and Pregnancy Conditions

Menstrual abnormalities have been recorded following both mRNA and adenovirus-based vaccinations, implying that, if there is a link, it is most likely due to the immunological response rather than a specific vaccine component; around a quarter of women infected with SARS-CoV-2 experience menstrual disruption [116]. A recent study concluded that vaccination is associated with a little increase in the cycle length but not the menses length [117]. Menstrual alterations have also been linked to typhoid, hepatitis B or human papillomavirus (HPV) vaccines; however, this is an issue that most women feel uncomfortable to discuss. Thus, the rates of such events may be somewhat higher [118]. Nevertheless, some experts suggest that the vaccine-induced thrombocytopenia is possibly related to heavy menstrual bleeding among women after vaccination, since blood loss can cause severe anemia, aggravate thrombocytopenia and increase the risk of hemorrhages and clots [118]. Genital necrosis with cutaneous thrombosis after COVID-19 mRNA vaccination was also reported in an 84-year-old woman with a history of DVT [119].

Conversely, mothers with a positive SARS-CoV-2 test result have a higher risk of abortion, as, during viral infection, placental inflammation can cause fetal development retardation and abortion [120]. Pregnant women were excluded from the initial clinical trials, but a study analyzing CDC data regarding pregnant women and mRNA vaccinations found that the risk of spontaneous abortion following mRNA vaccination, either before or during pregnancy, is comparable to the expected risk of spontaneous abortion; however, the study lacked a control group of unvaccinated pregnant women [121]. A study observed pre-eclampsia-like syndrome in few severe COVID-19 pneumonia cases, but, heretofore, there exist no data for this condition in vaccinated women [122]. Pre-eclampsia-like syndrome is related to Syncytin-1, but there is no cross-reactivity of anti-spike protein antibodies with this protein [123]. On the contrary, several observational data from pregnant women who received the mRNA-1273 vaccine during the second/third trimester have found no link between the vaccine and unfavorable pregnancy outcomes, and, while there is currently minimal information on the effects of immunization during the first trimester of pregnancy outcomes, no higher risk of miscarriage has been observed [23]. Evidence for placental transfer or milk excretion is still unavailable, but, on the contrary, mRNA-1273 vaccine advice highlights that the vaccination provider be informed of pregnancy or breastfeeding! [23]. The UK document presents 31,195 reproductive and breast disorders, including a death, and these reports counted >10,000 menstruation and uterine bleedings, >7000 menstruations with increased bleeding, >1000 breast signs/symptoms and some other events at lower rates [28].

#### 2.2.11. Muscle and Tissue Conditions

Myalgia and arthralgia are considered as very common adverse events after mRNA-1273 vaccination [23]. There exist reports for myopathy in the deltoid muscle with possible myositis, as well as for severe rhabdomyolysis without signs of myopathy or fasciitis 1 day after BNT162b2 vaccination; a 28-year-old healthy woman exhibited muscle soreness in her thigh muscles spreading to her lower legs and asymmetrical weakness in her lower limbs 5 days after receiving the BNT162b2 vaccine, and she was diagnosed with myalgia/paresis with rhabdomyolysis and fasciitis [124]. Although uncommon, macrophagic myofasciitis has been described following other past diseases’ vaccinations, and it has been linked to the aluminum hydroxide adjuvant used in some vaccines, but this adjuvant is not included in the current mRNA vaccine platforms [125]. COVID-19 can trigger polymyalgia rheumatica, and this condition was also reported following vaccination; Autoimmune/inflammatory Syndrome Induced by Adjuvants (ASIA syndrome) has been proposed by some researchers, and the authors highlight that the peripheral mononuclear blood cells of these patients have elevated TLR-7 and TLR-9 expression, and that the mRNA vaccination can also trigger innate immunity via these receptors [126]. Physicians must be aware of the likelihood of PMR development or recurrence in the context of universal COVID-19 vaccination, particularly in the first two weeks following vaccination [127].

Currently, there is no indication that mRNA vaccines are linked to an increase in arthritis flare-ups [128]. However, there is a report for a flare of rheumatoid arthritis in a middle-aged man at remission for 2 years, and the authors discuss that the exact mechanism of the flare is unknown, and that it is possible for a non-specific adjuvant effect to have occurred, or a possible molecular mimicry between the viral spike peptides and the patient’s self-peptides to have resulted in a flare; however, the likelihood that the flare’s timing in relation to the vaccination was coincidental cannot be excluded [129]. A case of Löfgren’s Syndrome after SARS-CoV-2 vaccination (second dose of the mRNA-1273 vaccine) is also evident, but this case had a first ChadOx-1 vaccine administration [130].

#### 2.2.12. Ear and Eye Conditions

Currently, according to VAERS, there have been more than 152 reports of tinnitus among 25,072 BNT162b2/mRNA-1273 vaccine recipients [131]. There have been three cases of sudden unilateral tinnitus after BNT162b2 vaccination reported that resolved rapidly, except for one case [132]. A middle-aged woman with PMHX of DMII and HTN developed throat fullness, wheezing, tinnitus and severe headache five minutes after BNT162b2 vaccination; she was treated and improved clinically, but her tinnitus has persisted for months despite treatments [131]. In the first reports, it was suggested that a hypersensitive reaction could be involved in the pathophysiology, resulting in an aberrant autoimmune response driven by circulating immune complexes or cytotoxic vestibule-cochlear autoantibodies, or a vasculitic event with localized cochlear destruction. In addition, patients’ pre-existing atopy and autoimmune illnesses may have raised the risk of an autoimmune response that was dysregulated [132]. Additionally, it could be related to an anxiety-related immunization reaction, as anxiety has been linked to severe/persistent tinnitus, while it cannot be excluded that an autoimmune inner ear condition or a coincidental event could also have occurred [132]. Tinnitus has rarely been reported after hepatitis B vaccination, and, also, anti-spike antibodies may react with antigens everywhere along the auditory pathway, causing inflammation in the tympanic membrane, ossicular chain, cochlea, cochlear arteries, organ of Corti and other parts of the auditory system [133].

A study found that 83.3% of the 30 patients with otologic conditions (18 of whom had an mRNA vaccination) had hearing loss, 50% had tinnitus, 26.7% had dizziness and 16.7% had vertigo; six had been previously diagnosed with Menière’s disease, two with Autoimmune Inner Ear Disease (AIED) and three with both conditions [134]. Vestibular neuritis was reported in a middle-aged man following BNT162b2 vaccination [135]. A recent systematic analysis of 28 case reports/series and 28 cross-sectional studies that met the criteria found a prevalence of 7.6% for hearing loss, 14.8 percent for tinnitus and 7.2 percent for rotatory vertigo, and another study revealed the incidence of audiovestibular symptoms—in particular, acute vertigo—and found that there was no difference in the type of nystagmus in patients subjected to BNT162b2 compared to the other [136]. The authors suggested that, because non-peripheral nystagmus is so common, a central nervous system involvement is a possibility, and an anxiety-mediated immune reaction was also proposed; however, current reports/studies are based on small sample sizes and do not report the difference between central or peripheral vertigo [136]. In addition, another study found an association of sudden sensorineural hearing loss (SSHL) with mRNA vaccination, but the study included a small size [137]. Indeed, sudden hearing loss, deafness, deafness unilateral, deafness neurosensory and hypoacusis were listed as an adverse event in the CDC VAERS, but a study based on data from this system found that no association between mRNA vaccination and SSHL really exists [138].

Regarding some eyes’ issues, uveitis was previously reported, along with sarcoidosis, but a study on few BNT162b2 recipients found that uveitis can occur both after the first and second doses or can exacerbate. However, most cases showed mild-to-moderate disease, and, also, after the second dose, one case of severe anterior uveitis developed vitritis and macular edema, which recovered entirely after an intravitreal dexamethasone injection [139].

There are few spontaneous ocular adverse events after mRNA-1237 vaccination, but there exists a report for a painless vision loss 48 h after the second vaccination and a subsequent hemispheric retinal vein occlusion diagnosis for a 74-year-old woman that was receiving oral anticoagulant therapy for atrial fibrillation [140]. Branch retinal arterial occlusion or combined arterial and venous occlusion and venous stasis retinopathy, as well as circumscribed nerve fiber infarction (with an onset of scotoma), have been some case series after both mRNA vaccinations in previously healthy and non-healthy individuals, and the short time interval between immunization and the development of the retinal vascular events increases the possibility of a causal relationship [141]. After receiving the first BNT162b vaccine administration, a middle-aged man with no previous ocular history developed bilateral acute painless blurring of vision, and Vogt-Koyanagi-Hara disease was diagnosed [142]. Moreover, a 34-year-old male was diagnosed with bilateral choroiditis one week after vaccination, and ultrasonography revealed a significant serous detachment of the macula in the right eye and substantial choroidal thickness in both eyes, but, eventually, the condition improved with a considerable resolution of the serous detachments and total visual recovery within two weeks after steroids’ treatment [143].

Moreover, a case series of numerical imbalance in incidences of Bell’s palsy between vaccine and placebo arms during trials, facial nerve palsy, a case report of acute-onset central serous retinopathy, panuveitis and a report of isolated abducens nerve palsy, as well as a report of penetrating keratoplasty patients with graft rejection, along with their related ocular symptoms, were summarized in the literature [144]. Solid organ transplant recipients were not included in the initial vaccines’ studies; thus, evidence is still reported, and there are few single-digit reports for acute unilateral/bilateral endothelial graft rejection after Descemet membrane endothelial keratoplasty or a few months after repeat penetrating keratoplasty that was originally performed years ago, central serous retinopathy with unilateral blurry vision and metamorphopsia, bilateral arteritic anterior ischemic optic neuropathy with a sudden loss of vision bilaterally and a case of bilateral acute zonal occult outer retinopathy with a progressive unilateral nasal defect and bilateral flashes [145]. The VAERS has received 46 reports of ocular side events generally associated with COVID-19 vaccination since February 2021, and the eyelid or conjunctiva were involved in the majority (74%), whereas the optic nerve, retina, uveal tract and anterior segment inflammation together contribute toward 9% [146]. Unilateral oculomotor nerve palsy with anti-GQ1b antibody following BNT162b2 vaccination is also uniquely reported, with diplopy, ptosis signs and a neurological examination revealing mild blepharoptosis, a limitation of adduction and vertical gaze [146]. Single cases of bilateral arteritic anterior ischemic optic neuropathy and bilateral acute zonal occult outer retinopathy were also reported, and the authors again concluded a possible autoimmune mechanism [147].

#### 2.2.13. Nervous System Conditions

An analysis study based on the adverse events that were reported in the VigiBase from 15 December 2020 to 24 January 2021 found that there were 19,529 reports related to clinical events and investigations about the neurological system, 15,638 events after BNT162b2 vaccination and 1075 events after mRNA-1273 vaccination [148]. In the analysis for the events following BNT162b2 vaccination, approximately 6/10 reported headache/dizziness, 7% reported paresthesia, 5% reported hypoesthesia, 3% reported presyncope/syncope, a high percentage reported somnolence and tremor, approximately 2.5% reported migraine, 2.5% reported dysgeusia, 2% reported lethargy and fewer than 1% reported anosmia/ageusia, balance disorders, burn sensation, dystasia, facial paralysis, hyperesthesia, loss of consciousness, neuralgia, aura, taste disorders, seizures, sensory disturbances/loss, speech disorders, stimuli unresponsiveness, ischemic stroke and memory impairment—also included were single-digit reports for some other rare events [148]. Fever resulting in status epilepticus was also reported following mRNA-1273 vaccination in an unconscious 56-year-old man with well-compensated secondary epilepsy [149]. Vocal cord paresis was also reported in a 28-year-old healthy woman following mRNA-1273 vaccination, and the authors discussed that this type of paresis was likely due to an inflammatory adverse reaction to the vaccine [150]. Nevertheless, a study analyzing VigiBase data for mRNA vaccines and facial paresis concluded that if there is a link between mRNA COVID-19 vaccinations and facial paralysis; the risk is very low, as in other viral vaccines [151].

We previously addressed a nationwide study that found no increase in the incidence of ischemic and hemorrhagic stroke 14 days after each BNT162b2 vaccination in adults aged 75 and older [87]. Aphasia and a following diagnosis of Intracerebral Bleeding (ICB) was also reported, 7 days after the second mRNA vaccination of a middle-aged man with a history of heart issues (post-myocardial infarction and stent implantation); it is known that ICB is rarely reported post-vaccination, yet the authors discuss that the vaccination may increase the systolic blood pressure or cause immune-mediated thrombocytopenia or hypocoagulability, since the 7-day latency excluded a potential immunological mechanism because T-cell immunization shows a 14-day window [152]. A study analyzing some serious neurological events occurring within a month after vaccination found that they occurred mainly after ChAdOx1 (85.5%), but fewer occurred after BNT162b2 (14.5%) vaccination, and none after mRNA-1273 vaccination [153]. Generally, Cerebral Venus Sinus Thrombosis (CVST) following SARS-CoV-2 vaccines is classified as either a CVST with vaccine-induced immune thrombotic thrombocytopenia syndrome (VITT), which is mostly associated with adenovirus vector-based vaccines, or a CVST without VITT characteristics, which is also associated with mRNA-based vaccines [154]. A study revealed that no case was observed with thrombocytopenia after mRNA vaccination, that cases after adenoviral vector vaccinations’ CVST had a worse prognosis and also that 38% (44/117) of the patients of the adenoviral vector vaccine group died, whereas 20% (2/10) of patients in the mRNA vaccine group died [155]. Some rare spontaneous events of intracerebral hemorrhage, delirium, myelitis, smell impairment, abducens nerve palsy, small fiber neuropathy and Parsonage–Turner syndrome were reported following mRNA vaccinations, and the authors concluded that some unexpected severe neurological conditions may have occurred as result of molecular mimicry and subsequent neuronal damage [155]. Acute encephalitis, myoclonus and Sweet syndrome were reported in a case that developed fever, generalized rash, confusion, orofacial movements and myoclonus after their first mRNA-1273 vaccination [156].

Demyelination was reported spontaneously and in multiples; the first report for Chronic Inflammatory Axonal Neuropathy (CIAN) describes an 82-year-old-woman who developed lower limb sensory loss and “pricking” associated with marked gait imbalance after her second BNT162b2 vaccination; the tests on Cerebrospinal Fluid (CSF) revealed GM3 antibodies and albumin-cytological dissociation, and, finally, the authors proposed a potential post-vaccination immune response [157]. A case series on mRNA vaccinations showed that seven patients experienced neurologic symptoms and MRI findings consistent with an active CNS demyelination of the optic nerve, brain and/or spinal cord within 3 weeks following vaccination, and visual loss, dysmetria, gait instability, paresthesia, sphincter disruption and limb weakness were among the symptoms; additionally, CSF showed oligoclonal IgG bands and MRI revealed lesions in certain cases [158]. The authors stated that these findings were anecdotal and that they do not indicate a link between SARS-CoV-2 mRNA vaccinations and active CNS demyelinating illness, whereas it is evident that rare cases of vaccination-associated demyelination have been spontaneously reported after vaccinations for influenza, hepatitis B, rabies, typhoid fever, smallpox, tetanus, polio and tuberculosis, such as Bell’s palsy, GBS, myelitis and MS [158]. New cases or relapses of Multiple Sclerosis (MS) were also recorded, and some authors suggest that vaccination and its adjuvants can cause autoimmunity, and that autoimmune reactions arise as a result of cross-reactivity with structurally identical host proteins [159]. A large study found that mRNA COVID-19 vaccines do not increase the short-term risk of MS clinical relapses [160]. Rare reports discuss Myelin Oligodendrocyte Glycoprotein (MOG) Antibody-associated Disease (MOGAD) (acute disseminated encephalomyelitis) after vaccination, and there exist some hypotheses about the postexposure synthesis of pathogenic IgG, as well as vaccine-induced bystander inflammation-induced tissue injury and antigen presentation (mimicry theory) or the emergence of a potential pre-existing autoimmune illness [161]. Furthermore, clinical trials did not report any Guillain–Barré Syndrome (GBS) occurrence following mRNA vaccinations, but there also exists a report for an old man with medical history who developed progressive bilateral lower limb weakness 20 days following BNT162b2 vaccination. The authors discussed a possible immune molecular mimicry, but, finally, they concluded that a temporal association is a possibility since there is still no molecular etiology for this event [162]. Another large study revealed that only 1/702 cases needed short medical care for a relapse of a previous syndrome, which represents a minimal risk for GBS after mRNA vaccination [163].

Despite the fact that two cases of severe dyskinesia after the BNT162b2 vaccine were reported, there is a converse case for a case with Parkinson’s Disease (PD) that improved following mRNA-1273 vaccination [163]. In addition, myasthenia Gravis was spontaneously recorded following vaccination, and some authors discuss the potential molecular mimicry and the bystander activation model—this may be more plausible [164].

Lastly, a few spontaneous steroid-responsive cases of aseptic meningitis following mRNA vaccinations were reported, and, particularly, there was a case of a heathy man with detected spike-specific IgG antibodies; some authors discussed that spike protein mimicry molecules created by vaccination may have broken the blood–brain barrier and caused aseptic meningitis, as well as a possible drug-induced aseptic meningitis (type III or type IV hypersensitivity reactions) [165].

#### 2.2.14. Psychiatric Conditions

Few spontaneous reports present Neuroleptic Malignant Syndrome (NMS) after vaccination; a 61-year-old woman with old schizophrenia and recent risperidone treatment developed high fever and disorientation a day after receiving the BNT162b2 vaccine, and, eventually, she was diagnosed with NMS [166]. Despite the fact that the event was a day after vaccination, the authors concluded that the vaccine had an immunostimulatory effect [166]. Acute mania with psychotic features was also described following mRNA vaccination; it manifested in a 42-year-old man through irritability, sleeplessness, delusions and a final amnesia of the whole situation after hospital discharge, as well as in a 57-year-old man with sleeplessness, irritability and a personal suicidal attempt [167]. The authors discussed that the vaccine-derived proteinic antigen can cause neuropsychiatric symptoms such as autoimmune psychosis; thus, risk groups should be carefully highlighted [167]. A 31-year-old man without previous history but with some schizotypal personality traits exhibited erratic and bizarre behavior 1 month after vaccination, but, finally, he was diagnosed with psychosis, and the authors suggested that the most logical explanation is that the vaccine, which has been shown to be safe in millions of people, may have induced psychotic symptoms in someone who was already vulnerable, most likely through a hyperinflammatory state [168]. There also exists a report about an 87-year-old woman with cancer history that developed anorexia, asthenia and vomiting 3 days after her second mRNA vaccination; she was found unconscious at home in deep hypoglycemia, and, finally, she was diagnosed with a central adrenal insufficiency, secondary to her cancer immunotherapy, as revealed by the vaccine-derived stress and immune reaction [169]. Agitation, disorientation to time, place and person and memory disturbances appeared in a 32-year-old man, since, initially, he was disoriented and amnesic 2 days after mRNA-1273 vaccination. He was diagnosed with acute hyperactive encephalopathy and a lack of efficacy [170]. Finally, there was another report for an 89-year-old man who had a delirious state 2 days after vaccination, and the authors discussed that this patient had many risk factors for delirium, including age, polypharmacy and sensitivity impairment, and they believed that the vaccine-induced immune response was sufficient in destabilizing his fragile homeostasis [171].

Figure 1 summarizes the diagnostic nature of the common spontaneous adverse events following the mRNA vaccination against COVID-19.

## 3. Evidence Regarding mRNA-LNP Based Strategies against COVID-19

Recommendations for mass vaccinations against COVID-19 and the so-called «Immune Passport» or «Vaccination Certificate» have been evident in populaces for more than a year now. At the same time, reports concerning post-vaccination adverse events have appeared during vaccination campaigns; several hypotheses have been made by experts for these occurrences, and diverse views have spread amongst societies.

First and foremost, it must be highlighted that the UK document, Yellow Cards, VAERS, VigiBase or related systems are undeniably based on sole suspicions of the reporters; thus, possible post-vaccination adverse events are not necessarily associated with vaccination. Previous misdiagnoses or undiagnosed medical conditions unrelated to vaccination can affect potential post-vaccination adverse reports. Reversely, post-vaccination adverse occurrences, even if healthcare providers are required to, could be under-reported or even not reported—if not monitored or if the physicians do not consider them as vaccine-related. Secondly, even if post-vaccination adverse events are more likely to be reported for the BNT162b2 vaccine, we must bear in mind that this was the first vaccine to receive emergency authorization; hence, more doses of this vaccine have been administered until now, and there is also a shorter interval between the first and second doses. Thus, reports might not reflect an honest situation. Undeniably, half-reporting (poor reports) and under-reporting based on half-diagnosis or under-diagnosis trammel the scientific community from making pure associations and assessments. The relative number, in combination with the reports’ nature, must not be used for the vaccines’ safety [28]. Additionally, adverse events’ occurrences in current literature have been recorded at a specific follow-up time and sole reports were presented until now; thus, further future assumptions are unclear. Furthermore, most related studies have small sample sizes—most reports even present sole cases; this phenomenon is linked to the limited observational period elapsed since vaccines were widely distributed, as well as the population’s inconsistent adherence to mass-vaccination [134]. The most complete data on adverse events were published in monitoring reports prepared by pharmaceuticals agencies or obtained through online questionnaires, and, as a result, such clinically unverified reports are subject to multiple and greater biases [134]. As a result, the initial aim of this review was to illustrate the nature of all current post-vaccination adverse events’ reports and related existent hypotheses, and, in this section, we further discuss some current common knowns and unknowns regarding mRNA vaccinations against COVID-19.

The WHO has defined five types of Adverse Events Following Immunization (AEFI) (reported in a 30-day-window after vaccination) that are discussed in the current literature and some spontaneous post-vaccination adverse events’ reports: (i) vaccine product-related reaction; (ii) vaccine quality defect-related reaction; (iii) immunization error-related reaction; (iv) immunization anxiety-related reaction; and (v) coincidental event [172]. Some types seem well-described, but, regarding the first and third type of probable adverse events in parallel with the mRNA vaccine technology, the current literature provides further data for: (i) LNP components, (ii) mRNA, and (iii) mRNA-produced spike proteins, without precluding their coexistence.

i.Generally, allergic reactions due to various vaccines have been revealed to occur at a rate of 1.31 cases per million vaccine doses, with no fatalities reported and no gender or age differences, despite the fact that slightly higher frequencies have been reported in women [173]. Hypersensitivity reactions after current vaccinations have been reported in approximately 1.5% of the recipients [149]. Generally, the confirmed unfavorable allergic reactions to vaccines are not attributed to the active bioingredients, but they are frequently related to the inactive ingredients/excipients. A pre-pandemic study on similar mRNA-LNP vaccines against H10N8 and H7N9 influenza viruses showed that the vaccine can distribute rather nonspecifically to several organs, such as the liver, spleen, heart, kidney, lung and brain, and, also, that the liver concentration is roughly 100 times lower than that of the intra-muscular injection site [174]. A study has shown that LNPs used in preclinical studies are highly inflammatory, by activating multiple inflammatory pathways and inducing IL-1β and IL-6, due to their ionizable lipid components [175]. Heretofore, no licensed vaccine includes the excipient Polyethylene Glycol (PEG), but this substance is non-toxic and was approved by the FDA in 1990, and it is found in a number of medications that have caused anaphylaxis, a potential fatal reaction. It has been proposed that people with pre-exposure to PEG and possible anti-PEG antibodies may be at risk for an allergic reaction due to current mRNA vaccines’ administrations. Literature data reveal that there is a wide variance in the measurements of pre-existing anti-PEG antibodies, ranging from 0.2% to 72% of healthy individuals [176]. Despite being thought of as safe and biologically inert, multiple studies have found that up to 70% of patients treated with PEGylated medicines generate anti-PEG IgG antibodies [177]. Nevertheless, some general reports on PEG reveal that the avidity of PEG-specific IgG from individuals with PEG anaphylaxis rose as the molecular weight of the PEG tested climbed from 1000 to above, with a clinical tolerance of PEG300 upon being challenged, implying that not all PEGs are equally dangerous at triggering responses; this may be attributed to various PEGs’ molecular weights [177]. While additional research is needed to understand the cause of the possible higher prevalence of allergy to mRNA vaccines compared to other vaccines, PEG 2000 is thought to be the most likely problem (before LNP enters into the cell, during it being inside the cell and after it is secreted by the cell) based on previous experience with PEG-conjugated biologics, yet it is labeled as a high-risk hidden allergen, as it is difficult to detect as a possible cause of allergic reaction [177,178]. IgE-mediated allergic reactions to PEGylated lipids have been reported, while Complement Activation-Related Pseudoallergy (CARPA) has also been seen in comparable liposomes; CARPA is partly attributed to the binding of pre-existing anti-PEG IgM to LNPs with a following complement activation [177,178]. Complement activation should occur in practically all vaccine recipients if LNP-based vaccines can elicit an immediate local complement activation, yet complement activation alone cannot explain anaphylactic episodes [178]. In addition, anaphylactic reactions to PEGylated nanomedicines such as pegnivacogin have been most common in people with high anti-PEG IgG titers, but not everyone with high levels of these antibodies had allergic reactions [179]. Moreover, mRNA vaccines contain sucrose, and not only PEG, as a stabilizing agent, and it is known that carbohydrate (sucrose) intolerance does not disturb the immune system but can cause common allergic reactions, namely gastrointestinal disturbances, as a stabilizer-induced anaphylaxis [179]. Additionally, the mRNA-1273 vaccine contains tromethamine (trometamol), which has been linked to allergy in people who have been exposed to gadolinium-based contrast media/iodinated contrast media, and, also, the excipient 1,2-distearoyl-sn-glycero-3- phosphocholine (DSPC) included in the LNPs of both mRNA vaccines could contribute to allergic reactions [34]. A non-related pre-pandemic study showed that that intravenously injected mRNA-LNPs can interact with phagocytic myeloid cells and neutrophils that can elevate cytokines—but determined that the PseudoU modification of mRNA does not reduce this immune response [180]. However, the extended biopersistence duration of such adjuvants or the ability of adjuvant particles to move and slowly accumulate in lymphoid organs or other tissues are major safety concerns. A non-related-to-COVID-19 study in mice showed that extracellular vesicles secreted after the endocytosis of that specific studied LNP-mRNA trigger inflammatory cytokines at lower rates than the original LNP, but the fate of the LNP-mRNA after endosome-engulfing and escape from the autophagy–lysosomal pathway is unclear [181].ii.Regarding the mRNA biomolecule, it has been said that extracellular RNA seems to predispose to endothelial damage, intercellular conjunctions relaxation and edema, increased viscosity, hyper-coagulation and thromboembolic events [182]. However, secreted RNases endowed with immuno-modulatory and antimicrobial properties can facilitate inflammation termination and tissue repair [183]. A pre-pandemic study showed that human fetal membranes respond to viral signatures in different ways, generating divergent inflammatory cytokine/chemokine profiles and antiviral responses [184]. This event, related to mRNA vaccination deriving and producing biomolecules, is currently unknown. The N1-methylpseudouridine reduces synthetic mRNA immunogenicity, but, concerning mRNA vaccinations against COVID-19, the relative influence of sequence engineering and m1Ψ incorporation of some specific immunogenic mechanisms into the cell remain to be reported [185]. Before translation, the mRNA could also bind to Pattern Recognition Receptors (PRRs) in endosomes or the cytosol prior in order to translate, and the final result is the activation of several pro-inflammatory cascades, such as inflammasome platforms, a type I interferon (IFN) response and the nuclear translocation of the transcription factor nuclear factor (NF)—kB [186]. The upregulation of these immunological pathways seems to be the basis for immune-mediated illnesses, particularly in genetically predisposed people who have an impaired clearance of nucleic acids, such as young women, due to the overexpression of X-linked genes because of the required antiviral response and the immune system’s stimulation by estrogens [186]. A recent study found that BNT162b2 is rapidly absorbed into the Huh7 human liver cell line in vitro, causing alterations in LINE-1 expression and distribution, and, after BNT162b2 exposure, BNT162b2 mRNA is reverse transcribed intracellularly into DNA in as little as 6 h; further research into the exact regulation of LINE-1 activity in response to BNT162b2 is needed [187]. Such findings raise questions on whether the vaccine-derived mRNA might be integrated to the human genome, resulting in potential genotoxic side effects, and, heretofore, it is unknown if DNA reversely transcribed from the mRNA vaccine is incorporated into the human cell genome in vivo [187].iii.Initially, the vaccine-derived mRNA is translated into proteins by ribosomes, used as an endogenous antigen and degraded by the proteasome into antigenic peptides, which are presented to CD8+ cytotoxic T cells via the MHC class I molecular pathway to activate cell-mediated immune responses; these products can be secreted into the extracellular space, thus entering the bloodstream, in which, they are uptaken by APCs, and antigenic peptides are delivered to CD4+ T lymphocytes through MHC class II molecules [188]. However, coronavirus infection was known for the massive production of viral proteins that can overload the folding capacities of the endoplasmic reticulum, and the possible misfolded antigens lead to final cell apoptosis, but this point is not studied regarding the vaccine-derived proteins [189]. A study showed that the circulating antigen was identified at detectable levels in the plasma of few mRNA vaccine recipients 1 day after vaccination. IgA levels were reported early, and IgG levels were reported 14 days post-vaccination; antigen levels declined and became undetectable by day 14 for most recipients, and no protein was detectable after the second vaccination, except for a case that was detected 1 day after but was undetected 2 days later [190]. On the contrary, a recent study revealed that mRNA vaccination stimulates robust germinal centers in lymph nodes containing vaccine mRNA and spike antigens up to 8 weeks post-vaccination in certain cases [191]. Some evidence indicates that the receptor-binding motif RBD, binding to human ACE2B (437–508), as well as to two fusion peptides, may be a possible allergen that may cause possible anaphylaxis, whereas the results from both tools point to the receptor-binding motif RBD of the spike protein as a probable allergen [192]. A study also assessing a mouse model injected with the S1 subunit spike protein concluded that the SARS-CoV-2 main pathology starts by ACE2 endothelial damage and that this pathology may be induced by the injected spike protein, and another study in mice revealed that the S1 protein crosses the Blood–Brain Barrier (BBB) [193,194]. The ACE2 receptors’ distribution in all tissues and women’s vulnerability due to chromosome X was evident from the very beginning of the COVID-19 pandemic; however, even if ACE2 mRNA is expressed homogeneously in all tissues, the same is not always the case for ACE2 protein expression [195]. A proteomic analysis revealed that SARS-CoV-2 proteins (including the spike protein) bind to hemoglobin and its metabolites, and that this—in extreme conditions—could even result in an integrated stress response and the global inhibition of mRNA translation, but, generally, in further medical conditions [196]. Furthermore, Human Endogenous Retroviruses (HERVs) or their viral products may enhance virus infection and penetration into human cells, or HERVs may create proteins that regulate the ribosome’s translation start, altering the pattern of COVID ORFs in various human hosts. In addition, a study showed that the spike protein induced in vitro human endogenous retroviruses’ HERV-W ENV expression, and, also, HERV-K was found to be elevated in COVID-19 patients [197]. The spike protein has also been linked with Alzheiner’s disease; like dementia and cognitive impairment, the Spike protein S1 RBD motif can bind to heparin and heparin-binding proteins, suggesting clues about neurodegeneration. In addition, the spike protein was found to have a significant binding to Toll-like Receptors (TLRs) in in silico studies—especially TLR4 [198,199,200]. Finally, two studies revealed a SARS-CoV-2 spike protein cross-reactivity and strong interactions with transglutaminases, myelin basic proteins, mitochondria, nuclear antigens, myosin, collagen, claudin5/6 and S100B, as well as possible interactions with many proteins, including proteins involved in synaptic vesicle trafficking, endocytosis, axonal transport, neuronal transmission, thrombosis, inflammation and the mitochondrial and blood–brain barrier, as well as protein growth factors [201,202].

The following Figure 2 summarizes the common current evidence regarding the mRNA related vaccination biomolecules.

## 4. Knowns and Unknowns: The Expert Opinion 

Two implausible and anecdotal undiscovered hypotheses would firstly be an mRNA undertranslation, consequent spike protein underproduction and further low immune response—a condition resembling a vaccine underdose—and, secondly, the reverse case, meaning a potential mRNA overtranslation and consequent spike protein overproduction under certain conditions, thus resulting in the overstimulation of the immune system, etc.—a condition resembling a vaccine overdose, under specific circumstances. These issues may sound unscientific, but remain unknown. Regarding the previously discussed study, related further concerns arise regarding the potential future transcription/translation of the incorporated sequences, their possible products and the potential immune responses. Such hypotheses would be extremely harmful in vulnerable cases with pre-existing medical conditions.

Based on the previous literature data, it seems strange and unscientific for some diagnosis of adverse events to be explained via produced antibodies’ cross-immunity shortly after vaccination, since there is evidence for their production within the window-day, and, also, some case reports do not even provide information regarding anti-spike protein antibody titers. Conversely, even if rare allergic reactions attributed to LNP usually happen some time following vaccination (anti-PGE antibody-mediated acceleration of LNP clearance and possible acute hypersensitivity), some studies have proved the LNP-mRNA existence for more than, i.e., the first week post-vaccination, and some reports have precluded this fact; however, the potential effect of current LNP degradation products seems to be unknown, and it is also unknown if, i.e., a spontaneous specific case’s cells show intolerance to LNP absorbance for a random reason, resulting in extracellular accumulation and further consequences. Nevertheless, almost all adverse reports discussing a potential LNP toxicity resulting in hypersensitivity reactions do not provide laboratory evidence but, rather, some clinical observations and hypotheses, and, consequently, these reports trammel the scientific community from identifying the causality of the possible allergic reaction—is the response due to LNP or the produced spike protein? Moreover, some reports try to explain post-vaccination autoimmune diseases’ occurrences, precluding that autoimmune conditions can be initially diagnosed or flare after vaccinations and infections too, and, additionally, no-one can know if someone already had a benign autoimmune disease that was not previously expressed. Furthermore, some reports lack an overall medical condition description, i.e., we detected only one case report with data for the post-vaccination type of menstrual bleeding and related alterations; undeniably, a puzzle has several pieces. In addition, we previously described a case report (skin adverse event) with a negative immunohistochemistry for the SARS-CoV-2 spike 1A9 protein, but the authors precluded that their used monoclonal antibody detected an S2 subunit, and literature data suggest much more than an S2 spike antigen; thus, even if this one was negative, we do not know the result for other spike antigens (even if a whole spike, or spliced, i.e., S1, RBD). Additionally, until now, no neurological case report has performed a nucleic acid/antigen detection test in CSF, and no respiratory case report has performed the same tests in BAL specimens; generally, heretofore, there is no related evidence for the vaccine’s biomolecules (test/biopsy) in current case reports besides the overall literature information—except some single-digit case reports, in total. We highlight that this evidence should be revealed in adverse reports and that accurate epidemiological studies exploring these biomolecules’ titers should be made, so as to identify more about their efficacy and to compare them with potential adverse events. However, one should bear in mind that current tests have been designed for SARS-CoV-2 and possible false tests have been explained, but we cannot know if all current test kits are able to detect the vaccine-related biomolecules—even if, theoretically, it sounds promising [6]. Additionally, the detection of a spike antigen in plasma samples can be hampered following the second vaccination due to the development of a circulating spike protein—anti-spike antibody immunological complexes. The reverse scenario for antibodies may exist, too [6]. Also, even if detected, several people passed COVID-19 thus they may be carriers of viral residues, and tests may be misleading, but may the overall biomolecules’ titers can explain some adverse events with the mRNA product likely to worsen them. Even in postmortem investigations of fatalities following vaccination, a causative association was suspected but still not established, but of course all sudden deaths should be disclosed and investigated so as to provide clear evidence in this post-vaccination era [203]. In addition, post-vaccination-reported COVID cases should be identified through test kits that detect different genes/antigens than the spike protein, and, also, tests identify SARS-CoV-2 carriers but not patients, so a positive test does not necessarily mean that the vaccine did not work/produce immunity, but also tests recognize carriers of at least genetic fragments thus the initial advice to disclose a COVID-19 infection with a positive test was unscientific and misleading [2,3,6,7]. The final suggestion is that more and precise laboratory diagnostic tests should be made, along with clinical evidence, so as to establish a correct causation for the various case reports, and they also should provide more and overall clinical and laboratory information regarding the past and the current medical condition of each case in order to conclude with accurate event explanations.

Nevertheless, and apart from the previous diagnostic analysis of the evidence regarding potential product-related AEFI against COVID-19, one could possibly doubt the «AEFI hypothesis» for the adverse events following an mRNA technology product administration in several ways. Firstly, literature has proven that LNP-mRNA can exist in humans for more than 30 days—at least a dual 2-month-window period thus it is unscientific to count adverse events due to vaccination just within a sole month (the long 3’ UTR of this mRNA is also evident). Moreover, the old vaccines’ theory was to insert mainly an inactivated virus, thus, adverse events of the product were mostly related to the ingredients of vaccine’s adjuvants; however, conversely, the mRNA technology (apart from the ingredients of the vaccine or adjuvants) inserts to the body the mRNA and a future active produced protein or proteinic biomolecules—the protein production works perfect. Additionally, if such a scenario for the potential mRNA integration into the human genome is real, then we are concerning ourselves with a long-lived product that does not fulfill the AEFI hypothesis; hence, further studies need to reveal information on this scenario.

Doubtlessly, the real pandemic rates cannot be assessed because not everyone had a real test result for SARS-CoV-2, but, reversely, the real mortality rate cannot be estimated since everyone with a positive test and a subsequent death was included in current mortality rates; hence current mortality rates are, in reality, lower—excess mortality rates are still being calculated. Undeniably, the initial advice to disclose deaths in such way was biased and misleading. As a result, vaccinations’ risks and benefits should be assessed more properly now, considering that we have lived with SARS-CoV-2 and its mutations for more than 2 years, and, also, considering that the initial emergency response to the pandemic has been overcome with the efficiency of the first vaccine doses. Furthermore, approximately ~3.5% of the human SARS-CoV-2 spike protein is still invariant [204]. As a result of the SARS-CoV-2 mutative potential as an RNA virus, along with the wild-type-based emergent vaccine development, there is a possibility of poor real-life results from mass vaccination and the survival of neutralization-resistant variants, demanding further treatment. To achieve optimal and long-term vaccination results, more well-designed and mutation-based studies are needed to identify epitopes with minimal heterogeneity. Despite the fact that there is no direct evidence that there is an Antibody-Dependent Enhancement (ADE) in COVID-19, this possible risk is also a huge challenge for prevention and vaccine development in the variants’ era [205]. Moreover, even if it seems to be an achievement to induce short-term circulating antibodies, thus leading to not having a viscous blood with further consequences, repeated booster doses can possibly lead to cumulative toxicity [190]. The overstimulation of innate immune responses may contribute to an excessive elicitation of inflammation and subsequent tissue injury. In addition, from a molecular biology perspective, protein translation is suppressed by the overstimulation of immune signaling; thus, the antigen expression is limited, with further consequences to vaccine efficacy. Thus, an annual vaccination against COVID-19 with safe vaccines, such as the influenza vaccine, could be a wiser scientific strategy.

One could speculate that «inflammation» seems to be the common denominator in several types’ reported diagnosis of post-vaccination spontaneous adverse events, and so steroids have been used in some certain case reports. The most crucial factor that scientific communities have to reassess is the real existence of spontaneous post-vaccination adverse events that do not result from the presence or exacerbation of existing comorbidities in certain cases, but this is extremely difficult to study, since some people may be already ill without knowing it (i.e., a benign or not yet expressed autoimmune condition) but, on the contrary, there already existed indications from pre-pandemic studies that doubt even the long-term safety of these vaccines. However, inflammation and treatments already exist in society, and the pre-existing medical conditions, including communicable and non-communicable diseases, were highly evident, giving vulnerability to several people, and, additionally, some populaces, such as Europe, are aging—undeniably, old age is a crucial comorbidity, thus several people should not receive this mRNA product as risks/benefits seem not balanced. In addition, further studies are taking place and are expected to prove the efficacy and safety of mRNA vaccines’ pediatric doses in younger children for a better protection against COVID-19 and to decrease the viral transmission rates. Evidence for mother-to-child vaccine-derived biomolecules’ transmission (breastfeeding) effects and safety are currently unknown regarding the newborn child, and, also, potential transmission via sex is also unexplored—however sudden related adverse events in unvaccinated children should be disclosed and investigated. Moreover, it was evident from pre-COVID-19 literature that after 1 h of mRNA-LNP injection, the production of the spike protein could be studied in liver cells—as a result, liver inflammation and related symptomatology should be thoroughly explored so as to prove that they are not vaccine-related adverse events. However, doubtlessly, the basic limitation of the current vaccine studies for adults and children is the longer-term safety information. The current literature highlights that certain people with underlying medical issues should not get vaccinated against COVID-19. However, even if there were an initial emergency need for prevention against COVID-19 and a short-term efficacy and safety against COVID-19 was successfully proven, the literature supports that more studies are required to prove the mRNA vaccines’ long-term safety; hence, more precautions should be taken right now, and recommendations for specific vaccinated people to check and monitor their health status should be evident.

In the last decades, therapeutics have been favorably revolutionized with nucleic acid platforms and technologies, and, particularly, LNP-mRNA-based technology has risen extremely quickly in the last few years, giving hope to several scientific innovations mainly towards medical issues. The so-called ‘RNA world’ has taught us that the valuable RNA molecule has both genetic and enzymatic properties, and one could hypothesize that we may have started with RNA—referring to life evolution—and we may end with RNA—referring to therapy evolution. The highly conserved antigen epitopes can be a sophisticated goal for mRNA vaccinations, and, also, such future vaccines could include conserved epitopes from other pathogens (such as influenza virus), so as to achieve an overall immunity for several pathogens’ mutants with multi-epitopic mRNA strategies; additionally, more adjuvants that stimulate innate immunity rather than hypersensitivity/inflammation should be designed. Nevertheless, we currently do not know the potential of the mRNA product to act as a ribozyme besides its initial role—this is based on the RNA world hypotheses. Doubtlessly, all of these suggestions need several thorough studies so as to prove these multi-epitope vaccines’ potential safety. However, it is required that scientific medical advances be in parallel with several morals, principally with «Primum, non Concere».

## 5. Conclusions

Current literature reveals several hypotheses regarding the spontaneous adverse reports following mRNA vaccinations against COVID-19, and there already existed some evidence concerning the mRNA technology before these vaccines’ appearance. Doubtlessly, precise diagnostic clinical and laboratory evidence and accurate case reports are required to prove some spontaneous associations, as well as to provide the mRNA technology’s safety for people since risk factors for adverse and severe conditions related to these vaccines are evident. However, current literature already reveals that several cases should not have received an mRNA vaccine against COVID-19 and should seek medical advice before potential next doses, thus highlighting a more personalized approach in the overall modern mRNA technology.

## Figures and Tables

**Figure 1 diagnostics-12-01555-f001:**
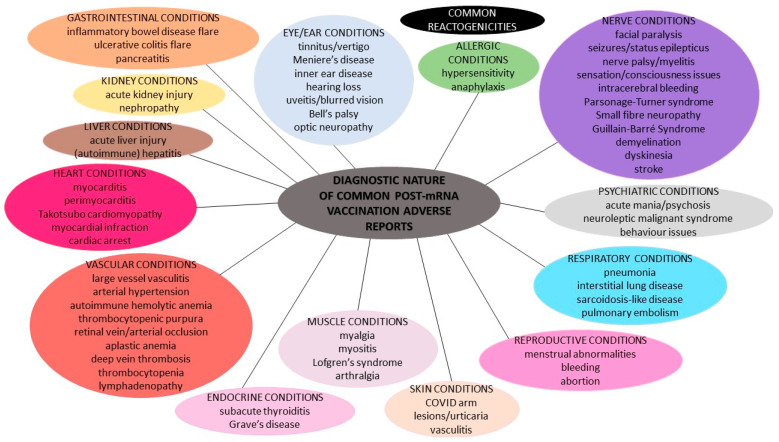
Diagnostic nature of the spontaneous adverse events following the mRNA vaccination against COVID-19.

**Figure 2 diagnostics-12-01555-f002:**
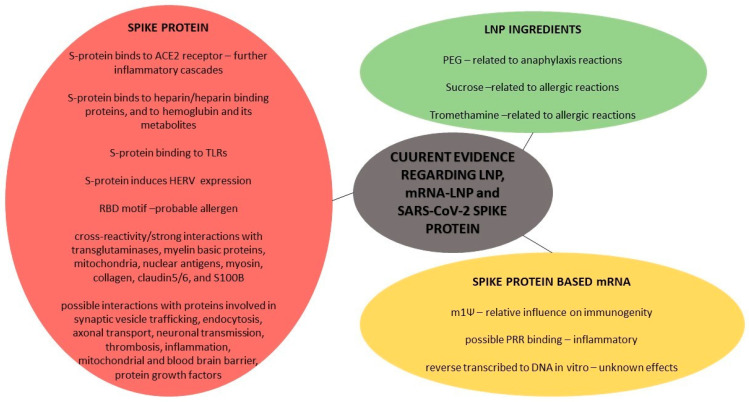
Summary of the current most common evidence regarding LNP, SARS-CoV-2 spike protein-based mRNA and spike protein.

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
