# Peer review of "Current Evidence in SARS-CoV-2 mRNA Vaccines and Post-Vaccination Adverse Reports: Knowns and Unknowns"

_diagnostics, 2022, doi:10.3390/diagnostics12071555_

Round 1
Reviewer 1 Report
Dear Authors,
Thank you for your submission. I enjoyed reading your manuscript. Overall, the review is far too long, especially the ”Knowns and Unknowns –Experts Opinion” chapter, but the information provided is up to date and useful. Maybe you could consider a shorter version.
Also, there are some minor errors in the text – for example line 1075 - without knowing in -- it
Author Response
Dear reviewer 1,
Thank you for your valuable time in our manuscript.
In reality, this state-of-the-art review article had the dual length, and this version has been submitted after several modifications and eliminations, thus it seems difficult for this paper to be again curtailed. Several points and typos have now been reviewed according to your wishes. Hope you now like this edited paper.

Reviewer 2 Report
During the two years of the COVID-19 pandemic, more than 519 million people were infected with the SARS-CoV-2 virus, and for 6.27 million patients, the infection ended in death. Almost 20% of infected people develop pneumonia and acute respiratory distress syndrome after a short period of time, which is the main cause of death in COVID-19. The main hope for ending the pandemic lies with the use of vaccines and the creation of herd immunity. To date, more than 30 vaccines have been approved and more than 340 are in preclinical or clinical trials. Many of these vaccines have passed phase III clinical trials and have shown a good safety profile. However, during their use, extensive information has also been accumulated about a wide variety of adverse events, many of which cannot always be directly associated with the vaccine. The aim of this review is to sum up the spontaneous adverse events following mRNA vaccinations. The review consists of 3 parts. The first part provides very complete information on all ever reported cases of complications after mRNA vaccines injection, including allergic, skin, vascular, hepatic reactions, as well as disfunctions of endocrine, respiratory, gastrointestinal, reproductive, nervous and urinary systems, psychiatric disorders and pathology of pregnancy, blood, ear, eye, heart, muscle and tissue.
The second part discusses in detail the side effects associated with vaccine mRNA accessory components (LNPs), the mRNA itself, and its protein product. Finally, the third part analyzes some of the unproven hypotheses regarding the occurrence of adverse events after vaccination with mRNA vaccines.
The review is written in a highly professional manner, is of undoubted interest to specialists and may be recommended for publication without any corrections.
Author Response
Dear reviewer 2,
Thank you for your valuable time in our manuscript and we really feel the need to thank you for your recommendation for publication in its current version. Thank you very much for your positive feedback. Best wishes.
